# *miR-194-3* Regulates Proliferation and Apoptosis of Follicular Granulosa Cells by Targeting *CHD4* in Zhedong White Geese

**DOI:** 10.3390/ani15182676

**Published:** 2025-09-12

**Authors:** Peng Zheng, Zhengyu Zang, Size Wang, Chuicheng Zeng, Yue Pan, Yuanliang Zhang, Shan Yue, Shengjun Liu, He Huang, Xiuhua Zhao

**Affiliations:** 1College of Animal Science and Technology, Northeast Agricultural University, Harbin 150030, China; zhengpeng@neau.edu.cn (P.Z.); zhengyuzang@126.com (Z.Z.);; 2Animal Husbandry Institute, Heilongjiang Academy of Agricultural Sciences, Harbin 150086, China; 3College of Animal Science and Veterinary Medicine, Heilongjiang Bayi Agricultural University, Daqing 163319, China

**Keywords:** goose, follicular granulosa cell, *miR-194-3*, *CHD4*, proliferation, apoptosis

## Abstract

Normal follicular development relies on a well-regulated balance between the proliferation and apoptosis of granulosa cells (GCs). This study investigated the role of *miR-194-3* in GCs of goose follicles. We found that *miR-194-3* was differentially expressed at various stages of follicular development and inhibited GC proliferation while promoting apoptosis by targeting *CHD4*. Further experiments revealed that GC proliferation was significantly enhanced and apoptosis was reduced when *CHD4* was silenced. These findings indicate that the *miR-194-3*–*CHD4* axis may contribute to the regulation of ovarian function in geese and provide a potential theoretical basis for further studies aimed at improving reproductive performance in poultry.

## 1. Introduction

Follicular development is a central process of female bird reproductive physiology, and its dynamic balance directly affects egg production performance. In avian ovaries, only a small number of primordial follicles develop into mature follicles and eventually enter the ovulation stage, while the majority of follicles degenerate due to atresia mechanism [1]. This process is closely related to the proliferation, differentiation and apoptosis of granulosa cells (GCs). As the main executors of follicular function, GCs coordinate follicular development and follicular maturation through the synthesis of hormones, growth factors and regulation of signaling pathways [2]. Studies have shown that abnormal apoptosis of GCs is considered to be one of the main causes of follicular atresia, which indirectly affects the avian egg-laying cycle [3]. Therefore, it is worthy to analyze the underlying regulatory mechanisms of this process to enhance poultry reproductive performance.

In recent years, microRNAs (miRNAs) have been found to be widely involved in the regulation of ovarian development and follicular function [4]. miRNAs modulate various cellular processes-including cell cycle progression, apoptosis, and differentiation, and bind to the 3′ untranslated region (3′UTR) of target mRNAs to suppress their expression [5]. In mammals, numerous miRNAs play crucial roles in GC function during follicular development [6]. The *miR-194* family comprises two mature sequences, *miR-194-5p* and *miR-194-3p* [7], they highly express in tissues such as liver, intestine, and genitourinary system [8,9,10]. These miRNAs are associated with process such as tumorigenesis [11], fibrosis [12], inflammatory [13], and more. *miR-194* expression upregulated in GCs from polycystic ovary syndrome (PCOS) patients. Overexpression of *miR-194* suppressed the proliferation of Human ovarian granulosa-like tumor cell line KGN cells by targeting Heparin-binding epidermal growth factor-like growth factor (*HB-EGF*) and promoted apoptosis [14]. In prostate cancer cells, *miR-194* inhibited cell survival and tumor growth by targeting N-calmodulin. Gao et al. further demonstrated that *miR-194* significantly reduced the proliferative capacity of these cells and induced apoptosis [15]. Collectively, these studies suggest that *miR-194* plays an inhibitory role in cell proliferation. However, there is still limited research on the role of *miR-194* in avian ovarian GCs, and a systematic study of its action and molecular mechanisms is needed.

Chromodomain helicase DNA-binding protein 4 (*CHD4*) is the core ATPase subunit of the Nucleosome Remodeling and Deacetylase (NuRD) chromatin remodeling complex. This complex regulates chromatin architecture and gene transcription by promoting ATP-dependent nucleosome sliding and histone deacetylation, thereby maintaining genomic integrity [16]. In mammalian ovaries, *CHD4* interacts with *MTA3* to form the NuRD complex, which regulates GC progression through the G2/M phase of the cell cycle, ultimately influencing *Cyclin B* expression and promoting cell proliferation [17]. To date, the role of *CHD4* or the NuRD complex in avian follicular development and GC proliferation or apoptosis has not been directly studied. However, transcriptomic analyses revealed differential gene expression patterns in chicken GCs at various developmental stages, suggesting that chromatin remodeling factors, including *CHD4*, may contribute to the regulation of follicular development [18]. The critical role of *CHD4* in chromatin structure remodeling and cell fate determination, requires further investigation into its mechanistic function in GCs.

In this study, we used GCs from Zhedong white geese to investigate whether miR-194-3 influences GC proliferation and apoptosis, and to explore potential molecular mediators underlying these processes. Guided by bioinformatic target prediction, we focused our analyses on CHD4 as a candidate target. To address these questions, we combined miRNA mimic/inhibitor experiments, siRNA-mediated gene knockdown, qRT-PCR, Western blotting, EdU incorporation, flow cytometry, and dual-luciferase reporter assays. These approaches were intended to clarify miR-194-3–dependent regulatory mechanisms that may contribute to follicular development in geese.

## 2. Materials and Methods

### 2.1. Laboratory Animals

The 18-month-old laying female geese used in the experiment were purchased from Zhejiang Xiangshan Wenjie White Goose Co., Ltd. (Xiangshan, Zhejiang, China) and randomly assigned to the wire mesh panel pen (0.5 m^2^/feather) for centralized rearing. Geese were provided with ad libitum access to feed and water using disk-type feeders (≥16 cm/goose) and dropper-type drinking nipples. Lighting and ventilation conditions were consistent with the local natural environment, without additional heating or artificial light supplementation. Ten 18-month-old Zhedong white geese were sampled during the egg-laying period, and another 10 were sampled during the brooding (nest-holding) period. Geese were first anesthetized with isoflurane inhalation, followed by euthanasia via intravenous injection of sodium pentobarbital after loss of consciousness. Gizzard, hepatic, cardiac, splenic, renal, pulmonary, and ovarian tissues were collected from the laying period, and ovarian tissues were collected from the brooding period (n = 10). For ovarian tissue sampling, ovaries were quickly excised, visible large hierarchical follicles (diameter >10 mm) were carefully removed before sampling to ensure comparability between geese and physiological stages. Approximately 100 mg of cortical tissue per goose was collected. 10 individual biological replicates were included for each group (laying and brooding). All samples were immediately snap-frozen in liquid nitrogen for further analysis.

### 2.2. GCs Isolation and Culture

Follicles were harvested from geese during the laying stage and were first classified into preovulatory follicles (F1–F5) and pre-hierarchical follicles (0–10 mm), which were classified into small white follicles (SWF), large white follicles (LWF), small yellow follicles (SYF), and large yellow follicles (LYF) based on size and color. All samples were preserved in liquid nitrogen. For all in vitro experiments, GCs were individually isolated from F1 follicles of different geese using the method described by Gilbert [19]. Cells from each goose were cultured and subjected to the same experimental procedures independently. The GCs were seeded into appropriate culture plates depending on the experimental requirements. Specifically, 6-well plates were used for transfection, RNA and protein extraction, and flow cytometry assays; 24-well plates were used for EdU cell proliferation assays; and 96-well plates were used for CCK-8 cell viability assays. Thus, unless otherwise stated, all experiments involving primary GCs were conducted using cells derived from F1-stage preovulatory follicles. All cells were maintained in Dulbecco’s Modified Eagle Medium/Nutrient Mixture F-12 (DMEM/F12, Hyclone, Logan, UT, USA) supplemented with 10% fetal bovine serum (Gibco, Carlsbad, CA, USA) and 1% penicillin–streptomycin (Gibco, Carlsbad, CA, USA), and incubated in a humidified atmosphere of 5% CO_2_ at 37 °C. Cell passaging and subsequent experiments were carried out when stable growth was achieved. All experimental procedures were approved by the Laboratory Animal Welfare Ethics Committee of Northeast Agricultural University (Approval No. SRM-06), and adhered to the Guidelines for Ethical Review of Laboratory Animal Welfare [20].

### 2.3. Cell Transfection

GCs were transfected when confluency reached approximately 80%. The *miR-194-3* mimic, inhibitor, and corresponding negative controls were designed based on the mature sequence of acyg-*miR-194-3* obtained from previous transcriptome analysis results [21]. *CHD4* targeting siRNA (si-*CHD4)* and siRNA negative control (si-NC) were designed based on the coding sequence of goose *CHD4* and specifically silences goose *CHD4* mRNA. All oligonucleotide sequences were synthesized by Beijing Sevin Innovative Biotechnology Co., Ltd. (Beijing, China) and Sangon Biotech Co., Ltd. (Shanghai, China), respectively. The *CHD4* gene sequence used in this study was obtained from the NCBI RefSeq database (accession number XM_067004496.1), based on the *Anser cygnoides* genome assembly GCF_000971095.1 (AnsCyg_PRJNA183603_v1.0). To investigate the targeting relationship between *miR-194-3* and *CHD4*, wild-type (WT) and mutant (MUT) sequences (200 bp upstream and downstream of the predicted *miR-194-3* binding site within the *CHD4* 3′UTR) were synthesized by Heilongjiang Genesoul Technology Co., Ltd. (Harbin, Heilongjiang, China). The sequences were listed in Table 1. Transfection of mimic, inhibitor, and siRNA was performed using Lipofectamine^®^ 2000 (Invitrogen, Carlsbad, CA, USA) according to the manufacturer’s instructions. We screened the concentrations of *miR-194-3* mimic and *miR-194-3* inhibitor, the three concentrations designed for mimic were 25 nM, 50 nM and 75 nM, and the three concentrations designed for inhibitor were 100 nM, 200 nM and 300 nM, respectively. We tested and screened the most suitable concentrations for transfection experiments. The optimal expression levels of *miR-194-3* mimic and *miR-194-3* inhibitor were 50 nM and 300 nM, respectively. All experiments were repeated three times. si-*CHD4* was performed at the recommended concentration of 50 nM.

### 2.4. RNA Extraction, Reverse Transcription and Fluorescence Quantitative PCR

Total RNA was extracted from Zhedong White goose tissues, and we transfected follicular GCs using a commercial RNA extraction reagent (Takara, Shiga, Japan). cDNA was synthesized using the stem-loop method with the miRNA cDNA synthesis kit (Gene-better, Beijing, China). cDNA was generated using the All-in-One First Strand cDNA Synthesis Kit II with dsDNase (Seven, Beijing, China) to analysis mRNA. Quantitative real-time PCR (qRT-PCR) was performed using the FastStart Universal SYBR Green Master (ROX) kit (Roche, Basel, Switzerland) in a 10 μL reaction volume, containing 5 μL of 2× SYBR Green Master Mix, 0.3 μL each of forward and reverse primers (final concentration: 300 nM), 1 μL of diluted cDNA template (approximately 20 ng), and 3.4 μL of RNase-free water. Each reaction was run in triplicate. U6 small nuclear RNA was used as the internal control for miRNA quantification, while GAPDH served as the internal control for mRNA expression analysis. qRT-PCR reactions were run on a QuantStudio™ 3 Real-Time PCR System (Thermo Fisher Scientific, Waltham, MA, USA), and data were analyzed using QuantStudio™ Design and Analysis Software v1.5.1 (Thermo Fisher Scientific, Waltham, MA, USA). All reactions were performed in triplicates. Relative gene expression were calculated using the 2^–ΔΔCT^ method. The specific primer sequences used in this study are listed in Table 2.

### 2.5. Target Gene Prediction

The target genes of *miR-194-3* were predicted using three bioinformatic databases: TargetScanHuman v8.0 (https://www.targetscan.org/, accessed on 24 October 2024), miRDB (http://mirdb.org/, accessed on 24 October 2024), and DIANA Tools (http://www.microrna.gr/, accessed on 24 October 2024). Only genes predicted by at least two databases were considered for further analysis.

### 2.6. Flow Cytometry

#### 2.6.1. Cell Cycle Analysis by Flow Cytometry

Cells were transfected with *miR-194-3* mimic, *miR-194-3* inhibitor, or si-*CHD4* in six-well plates and incubated for 48 h. Cells used for each biological replicate were isolated from a different individual Zhedong white goose. Then, the cells were harvested and fixed in 70% pre-chilled ethanol at 4 °C for 24 h. Finally, the cells were centrifuged to remove ethanol, washed with PBS, and stained with propidium iodide (PI) solution containing 50 μg/mL PI and 100 μg/mL RNase A (Beyotime Biotechnology, Shanghai, China) for 30 min at 37 °C in the dark. Each group included three biological replicates, and each replicate was analyzed in triplicate as technical replicates. Cell cycle distribution was analyzed by flow cytometry using a BD FACSCalibur™ cytometer (BD Biosciences, San Jose, CA, USA). Data acquisition and analysis were performed with FlowJo v10.0.7 (Tree Star Inc., Ashland, OR, USA).

#### 2.6.2. Flow Cytometry Analysis of Apoptosis

At 48 h post-transfection, cells were harvested and washed twice with cold PBS, then resuspended in 1× Binding Buffer (Meilun Biotechnology, Dalian, Liaoning, China) at a final concentration of 1 × 10^6^ cells/mL. A 100 µL aliquot of the cell suspension was transferred into a 5 mL flow cytometry tube. Subsequently, 5 µL of Annexin V-FITC and 5 µL of propidium iodide (PI) were added. The samples were gently vortexed and incubated for 15 min at room temperature in the dark. Then, 400 µL of Binding Buffer was added to each tube. Apoptotic were analyzed using a BD FACSCalibur™ flow cytometer (BD Biosciences, San Jose, CA, USA), and data were processed using FlowJo software v10.0.7 (Tree Star Inc., Ashland, OR, USA). Each group included three biological replicate. The apoptotic rate was calculated as the percentage of Annexin V^+^/PI^−^ (early apoptosis) and Annexin V^+^/PI^+^ (late apoptosis) cells among the total cell population, based on quadrant analysis in flow cytometry.

### 2.7. Western Blot

Total protein was extracted from follicular GCs of Zhedong White geese 72 h post-transfection using RIPA lysis buffer (Beyotime Biotechnology, Shanghai, China) supplemented with protease inhibitors. Protein samples were collected at 72 h post-transfection, as protein-level changes were more stable and prominent at this time point based on preliminary experiments, compared to earlier time points. Protein concentrations were determined using a BCA Protein Assay Kit (Beyotime Biotechnology, Shanghai, China), and equal amounts of protein samples (20 μg per lane) were mixed with 5× SDS loading buffer (Beyotime Biotechnology, Shanghai, China), boiled at 100 °C for 5 min, and stored at −20 °C until use. Proteins were separated by 12% SDS-PAGE and transferred onto PVDF membranes (Beyotime Biotechnology, Shanghai, China). Membranes were blocked with 5% non-fat milk in TBST for 2 h at room temperature, then incubated overnight at 4 °C with primary antibodies against *PCNA* (Catalog Number: AF6237, Affinity Biosciences, diluted 1:1000, Wuhan, China), *CDK2* (Catalog Number: AF6237, Affinity Biosciences, diluted 1:1000, Wuhan, China), *CCND1* (Catalog Number: AF0931, Affinity Biosciences, diluted 1:1000, Wuhan, China), *Bcl-2* (Catalog Number: AF6139, Affinity Biosciences, diluted 1:1000, Wuhan, China), *Caspase-3* (Catalog Number: WL04004, Wanleibio, diluted 1:500, Shenyang, China), *Caspase-9* (Catalog Number: WL01551, Wanleibio, diluted 1:1000, Shenyang, China), and *GAPDH* (Catalog Number: AF7021, Affinity Biosciences, diluted 1:3000, Wuhan, China). After washing three times in TBST (8 min each), membranes were incubated with HRP-conjugated goat anti-rabbit secondary antibody (1:5000; Proteintech Group, lnc.) for 2 h at room temperature. Signal detection was performed using enhanced chemiluminescence (ECL) reagents (Seven Biotrchnology, Beijing, China), and images were captured using a gel imaging system (Bio-Rad, Hercules, CA USA). Protein were quantified using ImageJ software (version 1.53; National Institutes of Health, Bethesda, MD, USA), with GAPDH as the loading control. Three biological replicates were included for each group.

### 2.8. Dual Luciferase Reporter Assay

Human embryonic kidney 293T (HEK293T) cells were used for the dual-luciferase reporter assay. Cells were co-transfected with the *miR-194-3* mimic and WT *CHD4* reporter plasmid, mimic NC and WT *CHD4* plasmid, *miR-194-3* mimic and MUT *CHD4* plasmid, or mimic NC and MUT *CHD4* plasmid in 6-well plates. After 48 h, cells were harvested, and luciferase activity was measured using a Dual-Luciferase^®^ Reporter Assay System (Beyotime Biotechnology, Shanghai, China) according to the manufacturer’s protocol. Each group was analyzed in three independent biological replicates, and each biological replicate was measured in triplicate as technical replicates.

### 2.9. CCK-8 Assay

GCs were seeded into 96-well plates and cultured for 24 h before transfection. Cell viability was assessed at 12, 24, 48 and 72 h post-transfection using the Cell Counting Kit-8 (CCK-8; Seven Biotechnology, Beijing, China). At each time point, 10 μL of CCK-8 reagent was added to each well 2 h prior to measurement. Absorbance was recorded at 450 nm using a microplate reader (BioTek Instruments, Winooski, VT, USA). Each group included three independent biological replicates, and each biological replicate was measured in triplicate as technical replicates.

### 2.10. EdU Assay

Zhedong White Goose follicular GCs were seeded into 24-well plates and cultured for 24 h before transfection. Then, cells were incubated with 500 μL of diluted EdU working solution per well for 2 h, following the instructions of the BeyoClick™ EdU-555 Cell Proliferation Detection Kit (Beyotime Biotechnology, Shanghai, China). The cells were then fixed, and the Click reaction solution—(comprising Click Reaction Buffer, CuSO_4_, Azide 555), and Click Additive Solution—was added and incubated for 30 min at room temperature in the dark. Subsequently, cell nuclei were stained with Hoechst 33342 (5 μg/mL) for 10 min at room temperature in the dark. After staining, cells were washed twice with PBS to remove excess dye prior to imaging. Fluorescence images were captured using a fluorescence microscope (Olympus IX73, Olympus, Tokyo, Japan). The proliferation rate was calculated by determining the ratio of EdU-positive cells to the total number of nuclei using ImageJ software (version 1.53; National Institutes of Health, Bethesda, MD, USA).

### 2.11. Data Analysis

All data were presented as mean ± SEM. Statistical analyses were performed using SPSS 20.0 software (IBM Corp., Armonk, NY, USA) and GraphPad Prism 8 (GraphPad Software, San Diego, CA, USA). For comparisons involving more than two groups, one-way analysis of variance (ANOVA) followed by Tukey’s post hoc test was used. For two-group comparisons (e.g., mimic vs. mimic NC or siRNA vs. si-NC), the unpaired two-tailed Student’s *t*-test was applied. All experiments were replicated a minimum of three times unless otherwise stated. Differences were considered statistically significant at *p* < 0.05 (* *p* < 0.05; ** *p* < 0.01).

## 3. Results

### 3.1. Ovarian Phenotypic Characteristic and Expression of miR-194-3 in Zhedong White Goose Tissues

Ovarian phenotypic characteristics during the laying period and brooding period were compared (Figure 1A,B). To investigate the expression of *miR-194-3* in gizzard, liver, heart, spleen, kidney, lung, and ovary of Zhedong white geese, qRT-PCR were performed. The results revealed that *miR-194-3* expression was significantly increased in ovarian tissue compared to the other tissues (Figure 1C, *p* < 0.01). Additionally, the expression of *miR-194-3* were quantitatively analyzed in follicles at different developmental stages, including SWF, LWF, SYF, LYF, F5, F4–F2, and F1. The expression of *miR-194-3* was significantly higher in F1 follicles than in follicles at other stages (Figure 1D, *p* < 0.01).

### 3.2. miR-194-3 Bioinformatic Analysis and Targeting CHD4

We first conducted a multi-species sequence alignment of *miR-194-3* and found it to be highly conserved (Figure 2A). To further explore the regulatory mechanisms of *miR-194-3* in Zhedong white goose follicle, we screened 34 potential target genes, including *CHD4*, *PTPN12*, *EPC2*, *FBXW7*, *CASK*, and *TRIP12*, using the bioinformatics tools TargetScan, miRDB, and DIANA (Figure 2B). To validate the authenticity of these predicted targets, we performed qRT-PCR after overexpressing and inhibiting *miR-194-3* in GCs. The expression of several predicted target genes: *CHD4*, *PTPN12*, *EPC2*, *FBXW7*, *CASK*, *TRIP12* changed significantly, decreasing upon *miR-194-3* overexpression and increasing upon inhibition (Figure 2C,D), suggesting a potential regulatory relationship. Among these, *CHD4* showed the most pronounced expression change (*p* < 0.01), indicating that it was a likely direct target of *miR-194-3*. Moreover, the expression of *miR-194-3* was significantly higher in ovarian tissue during the brooding period than during the laying period (Figure 2E, *p* < 0.05), whereas *CHD4* expression displayed the opposite trend (Figure 2F, *p* < 0.01). To confirm the direct interaction between *miR-194-3* and *CHD4*, we constructed wild-type and mutant dual-luciferase reporter plasmids (pmirGLO-*CHD4*-WT and pmirGLO-*CHD4*-MUT) containing the predicted *miR-194-3* binding sites (Figure 2I), and verified the insert sequences by Sanger sequencing (Figure 2G,H). Co-transfection of the *miR-194-3* mimic and pmirGLO-*CHD4*-WT into 293T cells significantly reduced luciferase activity compared to the mimic NC + pmirGLO-*CHD4*-WT group (Figure 2J, *p* < 0.01). However, no significant difference in luciferase activity was observed when *miR-194-3* mimic was co-transfected with pmirGLO-*CHD4*-MUT (Figure 2J).

### 3.3. The Inhibitory Effect of miR-194-3 on the Proliferation of Primary Follicular GCs in Zhedong White Geese

To investigate the effect of *miR-194-3* on the proliferation of primary follicular GCs in Zhedong white geese, we transfected cells with either a *miR-194-3* mimic or inhibitor. The relative mRNA expression of the proliferation marker genes *PCNA*, *CDK2*, and *CCND1* were measured by qRT-PCR. Additionally, cell proliferation was assessed using the CCK-8 and EdU assays, cell cycle was analyzed by flow cytometry, and relative protein expression of proliferation markers were evaluated by Western blotting. We first optimized the transfection concentration of the *miR-194-3* mimic by testing 25 nM, 50 nM, and 70 nM concentrations, and determined that 50 nM was the most effective (Figure 3A) (Figure 3A, *p* < 0.01). Overexpression of *miR-194-3* significantly decreased the mRNA expression of *PCNA*, *CDK2*, and *CCND1* (Figure 3B, *p* < 0.01). CCK-8 assay results showed a significant reduction in OD450 values at 48 h and 72 h post-transfection in the mimic group compared to the mimic NC group (Figure 3C, *p* < 0.01). Consistently, EdU staining revealed a marked decrease in the number of EdU-positive GCs in the mimic group (Figure 3D,F, *p* < 0.05). Flow cytometry showed that, after *miR-194-3* overexpression, the proportion of cells in G1 phase increased, while the proportion in S phase decreased, compared with the mimic NC group (Figure 3F,G, *p* < 0.01). Western blot analysis confirmed significant downregulation of *PCNA*, *CDK2*, and *CCND1* (Figure 3H,I, *p* < 0.05). Conversely, inhibition of *miR-194-3* significantly enhanced GC viability and proliferation. To determine the optimal concentration of the inhibitor, we tested 100 nM, 200 nM, and 300 nM, and found that 300 nM was the most effective concentration. (Figure 4A, *p* < 0.05). Following *miR-194-3* knockdown, qRT-PCR showed increased mRNA expression of *CDK2* and *CCND1* (Figure 4B, *p* < 0.05). The CCK-8 assay demonstrated significantly higher OD450 values at 48 h and 72 h post-transfection in the inhibitor group compared to the inhibitor NC group (Figure 4C, *p* < 0.01). Similarly, EdU assay results indicated a significant increase in EdU-positive GCs after *miR-194-3* knockdown (Figure 4D,E, *p* < 0.01). Flow cytometry showed that, in the inhibitor-treated group, the percentage of cells in S phase increased and the percentage in G1 phase decreased compared with the inhibitor NC group (Figure 4F,G, *p* < 0.01). Western blotting results further showed that inhibition of *miR-194-3* led to significantly increased protein levels of *PCNA*, *CDK2*, and *CCND1* (Figure 4H,I, *p* < 0.01).

### 3.4. The Promoting Effect of miR-194-3 on Apoptosis in Primary Follicular GCs of Zhedong White Geese

To investigate the effect of *miR-194-3* on apoptosis in primary follicular GCs of Zhedong white geese, we transfected cells with *miR-194-3* mimics and inhibitors. The relative mRNA expression of apoptosis-related marker genes (*Bcl-2*, *Caspase-3*, and *Caspase-9*) were analyzed by qRT-PCR. Apoptosis rates were evaluated by flow cytometry, and the corresponding protein expression were assessed by Western blotting. Overexpression of *miR-194-3* significantly downregulated *Bcl-2* mRNA expression while significantly upregulating *Caspase-3* and *Caspase-9* mRNA expression (Figure 5A, *p* < 0.05). Western blot analysis showed a consistent trend at the protein level (Figure 5B,C, *p* < 0.01). Flow cytometry revealed that the total apoptosis rate in the *miR-194-3* mimic group was significantly higher than in the mimic NC group (Figure 5D,E, *p* < 0.05). In contrast, inhibition of *miR-194-3* expression resulted in significantly increased *Bcl-2* mRNA levels and decreased *Caspase-3* and *Caspase-9* mRNA expression (Figure 5F, *p* < 0.01). Western blot analysis confirmed the same trend at the protein level (Figure 5G,H, *p* < 0.01). Furthermore, flow cytometry analysis showed a significantly lower apoptosis rate in the inhibitor group compared to the inhibitor NC group (Figure 5I,J, *p* < 0.05).

### 3.5. The Effect of CHD4 on Proliferation and Apoptosis of Primary Follicular GCs in Zhedong White Geese

To investigate the effect of *CHD4* on the proliferation and apoptosis of primary follicular GCs in Zhedong white geese, we performed a series of experiments using small interfering RNA targeting *CHD4* (si-*CHD4*). Knockdown of *CHD4* was achieved by transfecting si-*CHD4* into GCs. qRT-PCR analysis confirmed a significant reduction in *CHD4* mRNA expression following transfection (Figure 6A, *p* < 0.05). Furthermore, the mRNA levels of proliferation marker genes (*PCNA*, *CDK2*, and *CCND1*) were significantly downregulated (Figure 6B, *p* < 0.01). CCK-8 assays showed a marked decrease in cell viability at 72 h in the si-*CHD4* group compared to the si-NC group (Figure 6C, *p* < 0.01). EdU staining revealed a significantly lower number of proliferating GCs in the si-*CHD4* group (Figure 6D,E, *p* < 0.05). Cell cycle analysis demonstrated that knockdown of *CHD4* resulted in a pronounced G1-to-S phase transition arrest compared to the si-NC group via flow cytometry (Figure 6F,G, *p* < 0.05). Western blot analysis also confirmed decreased expression of *PCNA*, *CDK2*, and *CCND1* proteins (Figure 6H,I, *p* < 0.05). In contrast, qRT-PCR results showed that apoptosis-related genes *Caspase-3* and *Caspase-9* were significantly upregulated, while *Bcl-2* expression was significantly downregulated following *CHD4* knockdown (Figure 6L, *p* < 0.01). Consistently, Western blot results confirmed these trends at the protein level (Figure 6J,K, *p* < 0.05). Flow cytometry further revealed a significantly increased apoptosis rate in the si-*CHD4* group compared to the si-NC group (Figure 6M,N, *p* < 0.05).

## 4. Discussion

The precise balance between proliferation and apoptosis of ovarian GCs is essential for normal follicular development and is closely associated with the egg-laying performance of female birds [22]. Therefore, it is important to elucidate the mechanisms that regulate GC proliferation and apoptosis during follicle selection. Numerous studies demonstrated that microRNAs (miRNAs) were involved in follicular development [23]. In the present study, *miR-194-3* inhibited cell proliferation and promoted apoptosis, which was consistent with findings in various other cellular models. For example, *miR-194* suppresses proliferation and promoted apoptosis in esophageal squamous carcinoma cells by targeting KDM5B [24]. Similarly, *miR-194* overexpression arrested cell cycle progression and induced apoptosis in non-small cell lung cancer [25]. Moreover, *miR-194* promoted apoptosis by targeting *HB-EGF*, thereby inhibiting proliferation in GCs derived from patients with polycystic ovary syndrome (PCOS) [14]. These findings demonstrates that *miR-194-3* plays a conserved role in negatively regulating cell proliferation across different cell types. In our study, the expression of *miR-194-3* increased progressively during follicular development and was significantly upregulated at the F1 stage, suggesting that *miR-194-3* may play a key regulatory role in follicle maturation. Previous studies showed that miRNA expression was stage-specific during follicular development [26]. Our findings further reveal that *miR-194-3* exerts pro-apoptotic effects by upregulating *Caspase-3* and *Caspase-9*, and downregulating *Bcl-2*. In parallel, it suppresses proliferation by arresting the cell cycle, ultimately limiting follicular progression. The progressive increase in *miR-194-3* expression during follicular development suggests that it may be more relevant to later stages, rather than early follicle selection. It likely contributes to maintaining GC differentiation and tissue homeostasis post-dominance acquisition. However, the specific mechanism still needs further research.

A core ATPase subunit of the NuRD (nucleosome remodeling and deacetylase) chromatin remodeling complex is known to play a crucial role in DNA damage repair and cell cycle regulation [27]. Multiple studies demonstrated that pro-proliferative function of *CHD4* in modulating cell proliferation and apoptosis. For instance, Lin et al. reported that *CHD4* promoted cell proliferation and suppressed apoptosis in lung adenocarcinoma [28]. Similarly, D’Alesio et al. identified *CHD4* as a critical gene for breast cancer cell growth through RNA interference screening, with knockdown of *CHD4* markedly inhibiting tumor cell proliferation [29]. In addition, Kwintkiewicz et al. found that MAT3 could assemble with *CHD4* to form the NuRD complex in mouse ovaries, jointly regulating granulosa cell progression through the G2/M phase. Deletion of *CHD4* significantly reduced the expression of Cyclin B1/B2 and impaired granulosa cell proliferation [17], highlighting its important role in ovarian function. Our dual-luciferase reporter assay supports a direct interaction between *miR-194-3* and the *CHD4* 3′UTR in a heterologous reporter system. These results are consistent with the hypothesis that *miR-194-3* may exert part of its biological effects via *CHD4* suppression, but do not rule out contributions from additional targets or indirect downstream pathways. In the present study, knockdown of *CHD4* in goose GCs resulted in significant cell cycle arrest, along with the downregulation of proliferation-related genes and the upregulation of apoptosis-related genes. These findings reinforced the critical role of *CHD4* in promoting GC proliferation and were in strong agreement with previously reported results in mammalian GC models. Collectively, our findings suggest that *CHD4* promotes GCs proliferation and suppresses apoptosis in Zhedong white geese, at least in part by accelerating cell cycle progression. Furthermore, our in vivo expression data showed that *miR-194-3* was significantly upregulated in the ovaries during the brooding period, whereas *CHD4* expression was markedly decreased. This opposing expression pattern further supports a potential inverse regulatory relationship between the two molecules. The opposing expression patterns of *miR-194-3* and *CHD4* observed in vivo are consistent with a potential inverse regulatory relationship; together with the in vitro data, these observations suggest that the *miR-194-3*–*CHD4* interaction may contribute to follicular development in physiological contexts, but further in vivo validation is required.

We hypothesized that *miR-194-3* suppressed the expression of proliferation-associated proteins such as *PCNA*, *CDK2*, and *CCND1*, while it promoted the expression of pro-apoptotic factors including *Caspase-3* and *Caspase-9*, by downregulating *CHD4*. However, the antibody used in this study targeted total Caspase-3 rather than the cleaved (active) form. Since cleaved Caspase-3 is a more specific marker of apoptosis activation, future studies will incorporate cleaved Caspase-3-specific antibodies to more precisely confirm the involvement of the apoptotic pathway. This downregulation may impair the NuRD complex-mediated chromatin remodeling and reduce transcriptional activity of proliferation-related genes, ultimately resulting in cell cycle arrest and the activation of apoptotic pathways. Furthermore, the increased expression of *miR-194-3* observed in the ovaries of Zhedong White Geese during the clutching period may attenuate GC proliferation and promote follicular atresia via *CHD4* inhibition. This, in turn, may contribute to the cessation of egg-laying. These findings provided novel molecular insights into the regulation of ovarian function associated with broodiness in Zhedong White Geese.

Egg-laying performance in geese is significantly influenced by the number of developing follicles and the ovulation rhythm, with GC proliferation and survival directly determining the availability of functional follicles [30]. Our study demonstrated that inhibition of *miR-194-3* or upregulation of *CHD4* expression promoted GC proliferation and inhibited apoptosis, which may delay follicular atresia, thereby extending the laying period and enhancing egg production. In poultry production, it is conceivable that molecular regulation of *miR-194-3* expression could be used to optimize egg-laying performance and broodiness behavior in geese. Furthermore, given its high specificity in the ovary, *miR-194-3* may serve as a biomarker for predicting egg-laying potential. However, these applications require validation through extensive follow-up research. This study revealed regulatory pathway in the poultry reproductive system, enriching our understanding of non-coding RNA roles in follicle development. Future research should focus on identifying downstream effectors of this pathway and elucidate its connections with follicle selection, dominance, and other physiological processes, thereby laying a theoretical foundation for miRNA-based molecular breeding technologies.

Limitations and future directions: While our luciferase assay indicates a direct interaction between *miR-194-3* and the CHD4 3′UTR, this assay was performed in HEK293T cells and reflects binding in a heterologous reporter context. Although *CHD4* knockdown phenocopied several effects of *miR-194-3* overexpression, these data do not prove that *CHD4* is the sole mediator of *miR-194-3*′s actions in goose granulosa cells. Future experiments should include: (1) Rescue assays should be performed in primary granulosa cells by co-expressing a *CHD4* open reading frame (ORF) lacking its 3′UTR, so that it can be determined whether restoration of *CHD4* reverses the phenotypic effects induced by *miR-194-3*. (2) A minimum of two independent siRNAs targeting *CHD4* should be employed, and knockdown efficiency should be documented at both the mRNA and protein levels. (3) Apoptotic activation should be confirmed through measurement of cleaved (active) apoptotic markers, for example, cleaved caspase-3, in order to verify execution of the apoptotic program. (4) Correlation analyses and functional assays conducted in vivo should be undertaken to reinforce the physiological relevance of the in vitro findings.

## 5. Conclusions

In this study, we provide evidence that *CHD4* is a direct target of *miR-194-3* and show that *miR-194-3* expression and *CHD4* knockdown produce similar effects on GC proliferation and apoptosis. These results support the view that *miR-194-3* can influence GC behavior at least in part via *CHD4* downregulation, but further mechanistic and rescue experiments are required to establish the extent to which *CHD4* mediates these effects. Notably, *miR-194-3* was highly expressed in GCs of F1 follicles, suggesting that the *miR-194-3*/*CHD4* axis may primarily function at the late stage of follicular development. This regulatory pathway may affect the process of follicular atresia by modulating the balance between GC proliferation and apoptosis.

## Figures and Tables

**Figure 1 animals-15-02676-f001:**
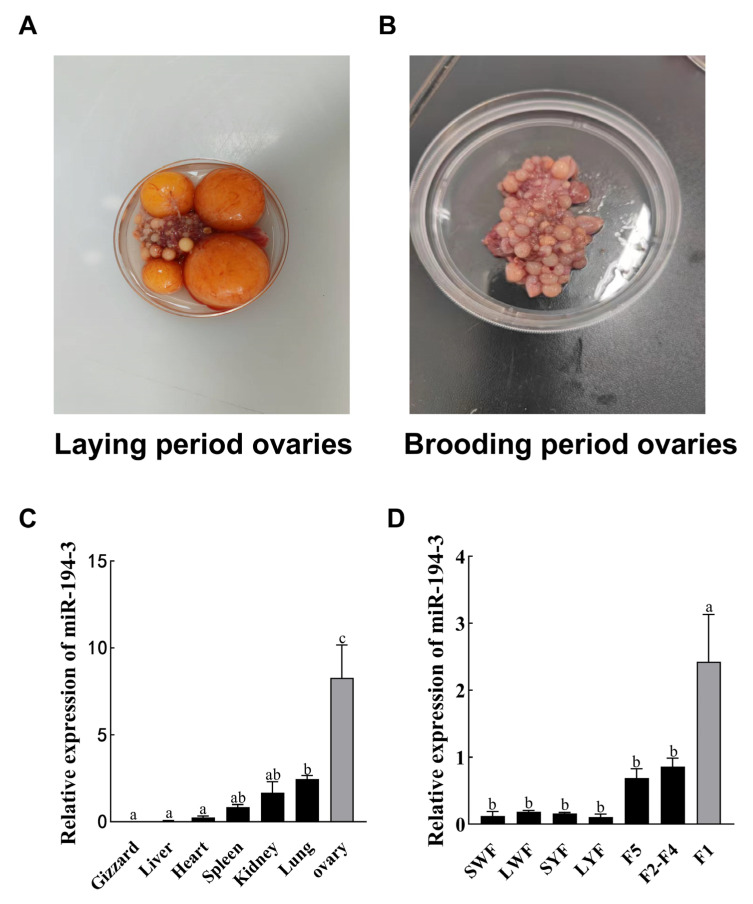
Expression of *miR-194-3* in Zhedong white goose. (**A**,**B**) Ovarian phenotypic characteristic during laying and brooding periods. (**C**) *miR-194-3* expression in various tissues of Zhedong white goose. (**D**) *miR-194-3* expression in follicles of Zhedong white geese at various stages. Data are expressed as mean ± SEM (n = 10). Different letters (a, b, c) denote means that are significantly different from each other (Tukey’s post hoc test, *p* < 0.05).

**Figure 2 animals-15-02676-f002:**
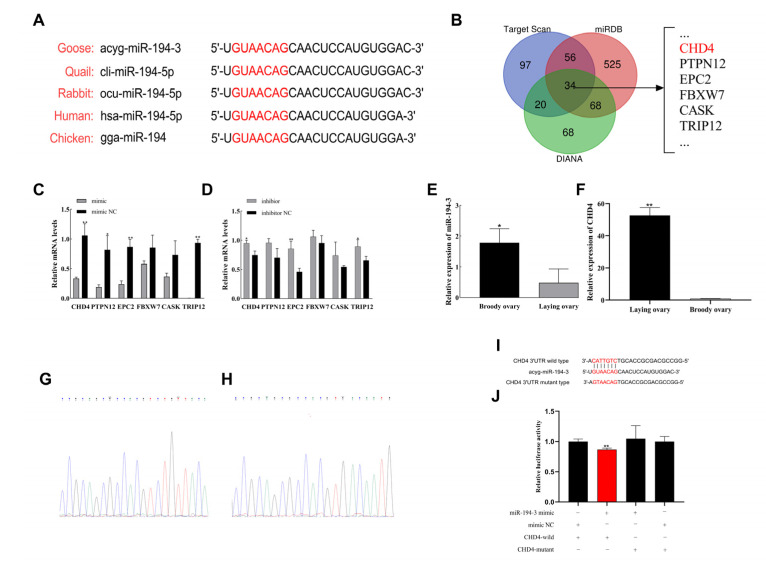
Chromodomain Helicase DNA Binding Protein 4 (*CHD4*) is a target gene of *miR-194-3*. (**A**) Mature sequences of *miR-194-3* from different species. (**B**) Venn diagram of *miR-194-3* target genes screened according to three websites, miRDB, Target Scan, and DIANA. (**C**,**D**) qPT-PCR detection of target gene expression after overexpression and knockdown of *miR-194-3*. (**E**,**F**) qPT-PCR detection of *miR-194-3* and *CHD4* expression in ovaries during laying and clutching. (**G**,**H**) Sequencing results of *CHD4* wild-type and mutant plasmids. (**I**) The target position of *miR-194-3* seed sequence on the mRNA of *CHD4* was predicted by TargetScanhuman v8.0. (**J**) pmirGLO-*CHD4*-WT, pmirGLO-*CHD4*-MUT and *miR-194-3* mimic, mimic NC were co-transfected into 293T cells, respectively, and luciferase activity was measured after 48 h. Data are expressed as mean ± SEM (n = 3). * *p* < 0.05, ** *p* < 0.01. Red letters indicate the seed sequence of *miR-194-3* (Panel A) or mutated nucleotides in *CHD4* 3′UTR (Panel I).

**Figure 3 animals-15-02676-f003:**
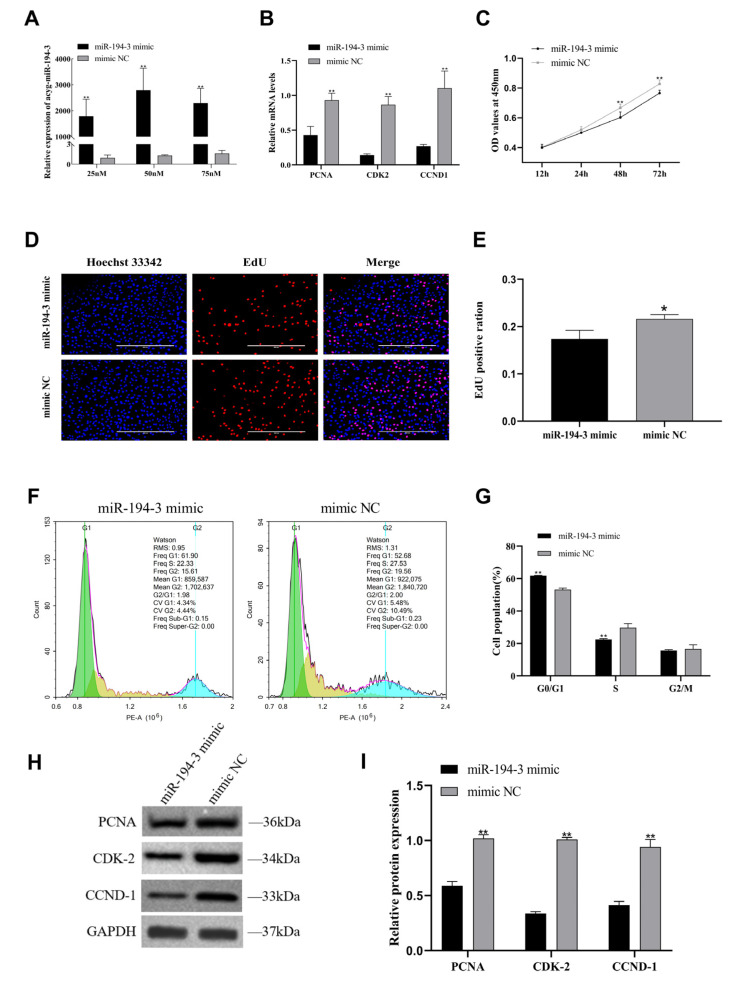
*miR-194-3* mimic regulates the proliferation of follicular GCs in Zhedong white geese. (**A**) Relative expression of *miR-194-3* after transfection with different concentrations of *miR-194-3* mimic (25 nM, 50 nM, and 70 nM), as measured by RT-qPCR. (**B**) After *miR-194-3* mimic and mimic NC transfection of GCs, qPT-PCR was performed to detect the expression of proliferation marker genes *PCNA*, *CDK-2* and *CCND-1* mRNA. (**C**) Cell viability was determined by CCK-8 assay after *miR-194-3* mimic and mimic NC transfection of GCs. (**D**) Representative images of EdU staining in GCs transfected with *miR-194-3* mimic or mimic NC. Nuclei were stained with Hoechst (blue), and proliferating (EdU-positive) cells are shown in red. (**E**) Quantification of the EdU-positive cell ratio in each group. Data are presented as mean ± SEM from three independent experiments. * *p* < 0.05 vs. mimic NC group. (**F**,**G**) Flow cytometry detection of granulocyte cell cycle profile 48 h after *miR-194-3* mimic and mimic NC transfection of granulocytes. (**H**,**I**) Western blotting results of *PCNA, CDK-2*, *CCND-1* and relative protein expression of *miR-194-3* mimic and mimic NC after transfection of GCs for 72 h. Data are expressed as mean ± SEM (n = 3). * *p* < 0.05, ** *p* < 0.01.

**Figure 4 animals-15-02676-f004:**
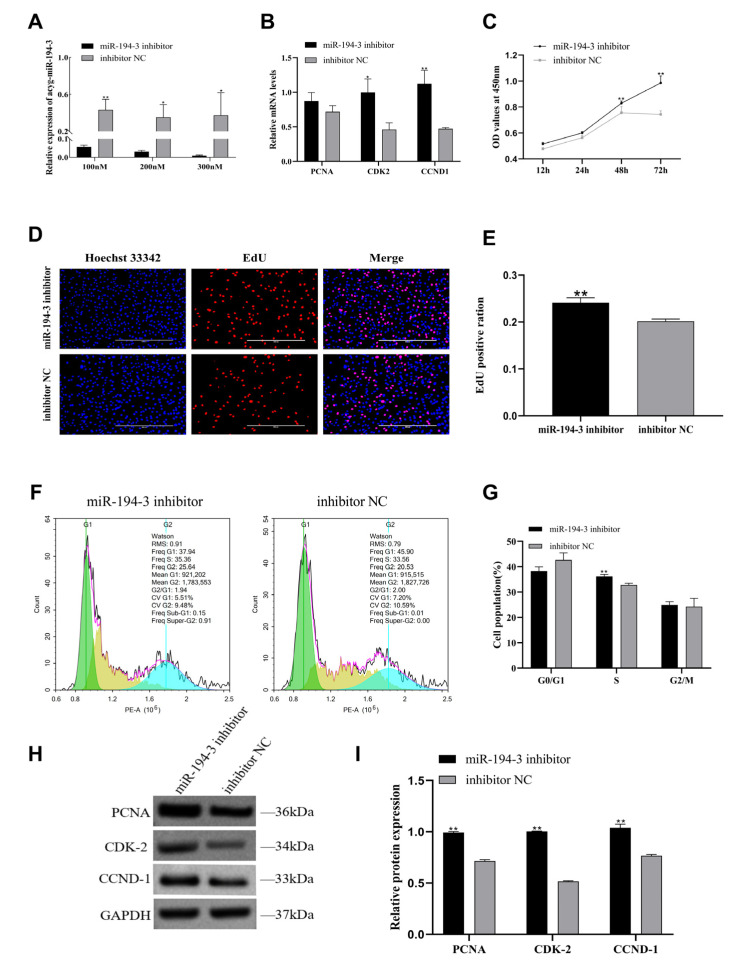
*miR-194-3* inhibitor regulates the proliferation of follicular GCs in Zhedong white geese. (**A**) Relative expression of *miR-194-3* after transfection with different concentrations of *miR-194-3* inhibitor (100 nM, 200 nM, and 300 nM), as measured by RT-qPCR. (**B**) After transfection of GCs with *miR-194-3* inhibitor and inhibitor NC, qPT-PCR was performed to detect the expression of proliferation marker genes *PCNA*, *CDK-2* and *CCND-1* mRNA. (**C**) Cell viability was determined by CCK-8 assay after *miR-194-3* inhibitor and inhibitor NC transfection of GCs. (**D**,**E**) Positive granulocytes (red) and Hoechst-stained total cells (blue) were detected by EdU assay 48 h after *miR-194-3* inhibitor and inhibitor NC transfection of granulocytes. (**F**,**G**) Flow cytometry detection of granulocyte cell cycle after *miR-194-3* inhibitor and inhibitor NC transfection of granulocytes for 48 h. (**H**,**I**) Western blotting results of *PCNA*, *CDK-2*, *CCND-1* and relative protein expression of *miR-194-3* inhibitor and inhibitor NC 72 h after transfection of GCs. Data are expressed as mean ± SEM (n = 3). * *p* < 0.05, ** *p* < 0.01.

**Figure 5 animals-15-02676-f005:**
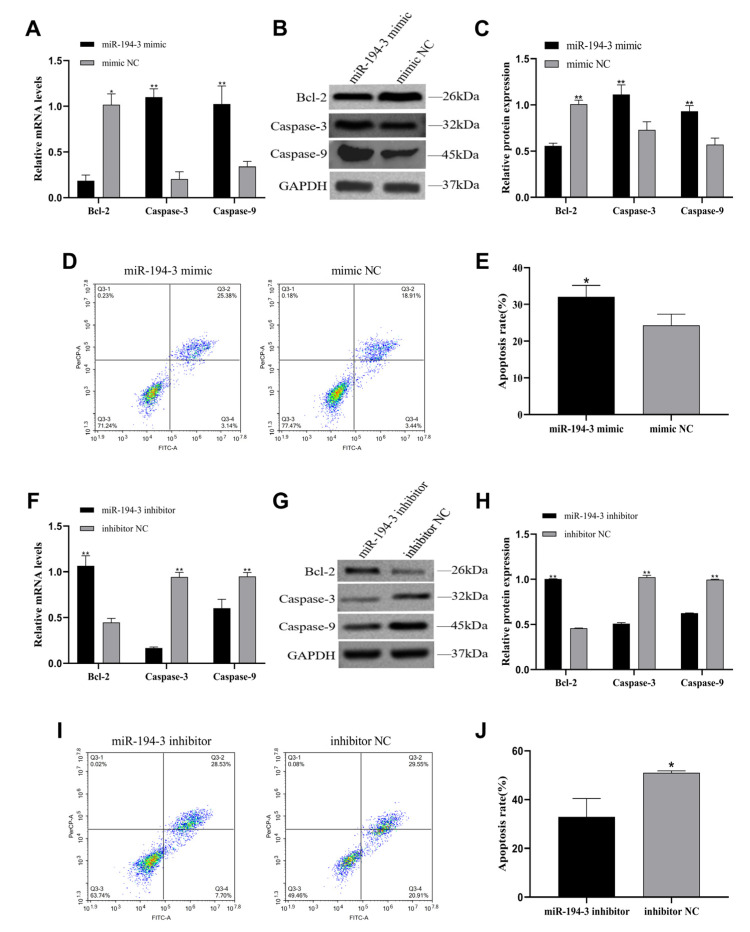
*miR-194-3* regulates apoptosis in follicular GCs of Zhedong white goose. (**A**) After *miR-194-3* mimic and mimic NC transfection of GCs, qPT-PCR was performed to detect the expression of apoptosis marker genes *Bcl-2*, *Caspase-3* and *Caspase-9* mRNA. (**B**,**C**) Western blotting results of *Bcl-2*, *Caspase-3*, *Caspase-9* and relative protein expression of *miR-194-3* mimic and mimic NC after transfection of granulosa cells for 72 h. (**D**,**E**) Flow cytometry detection of apoptosis and the proportion of total apoptotic cells 48 h after *miR-194-3* mimic and mimic NC transfection of granulocytes. (**F**) After *miR-194-3* inhibitor and inhibitor NC transfection of GCs, qPT-PCR was performed to detect the expression of apoptosis marker genes *Bcl-2*, *Caspase-3* and *Caspase-9* mRNA. (**G**,**H**) Western blotting results of *Bcl-2*, *Caspase-3*, *Caspase-9* and relative protein expression of *miR-194-3* inhibitor and inhibitor NC after transfection of GCs for 72 h. (**I**,**J**) Flow cytometry detection of apoptosis and the proportion of total apoptotic cells 48 h after *miR-194-3* inhibitor and inhibitor NC transfection of granulocytes. Data are expressed as mean ± SEM (n = 3). * *p* < 0.05, ** *p* < 0.01.

**Figure 6 animals-15-02676-f006:**
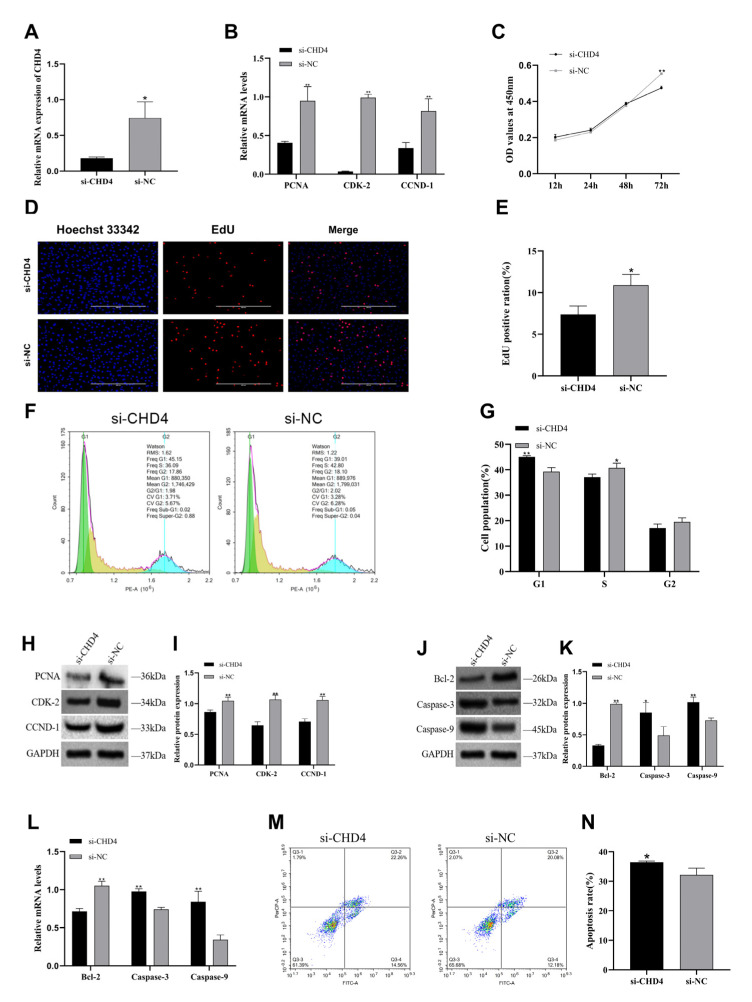
*CHD4* regulates proliferation and apoptosis of follicular GCs in Zhedong white geese. (**A**) *CHD4* mRNA expression after transfection of si-*CHD4* and si-NC into GCs. (**B**) After transfection of si-*CHD4* and its NC into GCs, qPT-PCR was performed to detect the mRNA expression of proliferation marker genes *PCNA*, *CDK-2*, and *CCND-1*. (**C**) Cell viability was determined by CCK-8 assay after transfection of si-*CHD4* and its NC into GCs. (**D**,**E**) Positive granulocytes (red) and Hoechst-stained total cells (blue) were detected by EdU assay 72 h after transfection of granulocytes with si-*CHD4* and its NC. (**F**,**G**) Flow cytometry detection of granulocyte cell cycle profile 72 h after transfection of granulocytes with si-*CHD4* and its NC. (**H**,**I**) Western blotting results of *PCNA*, *CDK-2*, *CCND-1* and relative protein expression of si-*CHD4* and its NC after transfection of GCs for 72 h. (**J**,**K**) Western blotting results of *Bcl-2*, *Caspase-3*, *Caspase-9* and relative protein expression of si-*CHD4* and its NC after transfection of GCs for 72 h. (**L**) qPT-PCR detection of apoptosis marker genes *Bcl-2*, *Caspase-3*, *Caspase-9* mRNA expression after transfection of si-*CHD4* and its NC into GCs. (**M**,**N**) After 72 h of transfection of si-*CHD4* and its NC into GCs, apoptosis and the proportion of total apoptotic cells were detected by flow cytometry. Data are expressed as mean ± SEM (n = 3). * *p* < 0.05, ** *p* < 0.01.

**Table 1 animals-15-02676-t001:** RNA oligonucleotide sequence.

Gene Name	Gene Sequence (5′-3′)
acyg-*miR-194-3* mimic	UGUAACAGCAACUCCAUGUGGAC
mimic NC	UUGUACUACACAAAAGUACUG
acyg-*miR-194-3* inhibitor	GUCCCACAUGGAGUUGCUGUUACA
Inhibitor NC	CAGUACUUUUGUGUAGUACAA
si-*CHD4*	AGAUGGAGAUUCUGUUGAATT
siRNA-NC	UUCUCCGAACGUGUCACGUTT

**Table 2 animals-15-02676-t002:** qRT-PCR primer sequences.

Gene Name	Sequence (5′-3′)	Product Length (bp)
*PCNA*	F: TGTTCCTCTGGTTGTGGAGTA	90
	R: GAGCCTTCTTGTTGGTCTTCA	
*CDK-2*	F: CTCCACCTCCAAGTTCCTAATG	89
	R: GCTGATCTATGGCACTGTCC	
*CCND-1*	F: TTCATCGCCCTTTGTGCC	80
	R: ATTGCTCCCACGCTTCCA	
*Bcl-2*	F:GATGCCTTCGTGGAGTTGTATG	100
	R: GCTCCCACCAGAACCAAAC	
*Caspase-3*	F: CTGGTATTGAGGCAGACAGTGG	158
	R: CAGCACCCTACACAGAGACTGAA	
*Caspase-9*	F: GTCCAAGACCAGAGCGAACA	122
	R: ATCAGGCAGTGTCCACAAGG	
*CHD4*	F: TGAAGAGGAGATGGGGGAGG	133
	R: ATTCCTGGCCAGATCCTCCT	
*PTPN12*	F: TGCCGAAGCCAGTTGTGATTG	116
	R: ACGACCAGGTAGTACAGGTGAAG	
*EPC2*	F: AACCGCTGCCCATCTACCG	115
	R: TGTTCCGATTCCTCCTCCTTCTC	
*FBXW7*	F: CTGATGACAGCAGTAGAGAAGATGAG	102
	R: TGTAGAATGGTGATGACTGGTGAATG	
*CASK*	F: GCGTTGTGCGGCGATGTATC	104
	R: ATCTTCTGTGCTTAATCCAGGACTTG	
*TRIP12*	F: TTATCTGTGAACTGATGCCTTGTCTG	115
	R: TCATCTCGCCATTGCCATATTGC	
*GAPDH*	F: TTTCCCCACAGCCTTAGCA	90
	R: GCCATCACAGCCACACAGA	
*miR-194-3*	F: ACGGAACTGTAACAGCAACTCCA	
	R: ATCCAGTGCAGGGTCCGAGG	
*miR-194-3* RT primer	GTCGTATCCAGTGCAGGGTCCGAGGTATTCGCACTGGATACGACGTCCAC	
U6	F: ATTGGAACGATACAGAGAAGATT	
	R: GGAACGCTTCACGAATTTG	

## Data Availability

Original experimental data in the Appendix A of this study are available from the authors upon request.

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
