# Peer review of "miR-194-3 Regulates Proliferation and Apoptosis of Follicular Granulosa Cells by Targeting CHD4 in Zhedong White Geese"

_animals, 2025, doi:10.3390/ani15182676_

Round 1

Reviewer 1 Report

Comments and Suggestions for Authors

The study focuses on the role of miR-194-3 and CHD4 in granulosa cells of laying geese and their effects on cell proliferation and apoptosis. Through a series of experiments, the authors provide evidence suggesting that miR-194-3 decreases cell proliferation and apoptosis and targets CDH4. Silencing CDH4 produced similar results. This study provides insights into a mechanism for granulosa cell proliferation and apoptosis, a process necessary for efficient follicle development. Ultimately, this research could provide important background to improve reproductive efficiency in avian species.

While the scientific premise and potential implications of this study are compelling, the manuscript requires significant revisions in terms of organization and clarity. The method section lacks sufficient detail to evaluate the reproducibility of the experiments. The cell culture experiment designs are not clear, making it difficult to understand the cell culture conditions and timing of incubations following various transfections. The justification for the use of granulosa cells from F5 follicles geese in the laying stage should be included, since miR-194-3 expression was shown to be more highly expressed in geese in the brooding stage. Additionally, showing the expression of CHD4 across follicle development could strengthen the data and help justify the selection of granulosa cells at a specific size for further analysis. There are also discrepancies in the statistical methods description as the only analysis described is a one-way ANOVA with post hoc Tukey test. Several analyses were performed using less than 3 experimental groups, for which an ANOVA would not be appropriate. A few analyses are shown in the results section, but are not included in the methods section, making it difficult to evaluate. Overall, the methods section should include more detail to improve readability and ensure reproducibility.

The results section does not describe the data shown in the figures in sufficient detail, making it difficult for the reader to understand the outcomes of the experiments. The authors include interpretative statements throughout the results section which would be more appropriately placed in the discussion. The discussion provides a solid overview of the literature related to CHD4 and miR-194-3, but it would be strengthened by incorporating the authors’ interpretations from the results section and more clearly contextualizing the findings within the current literature.

While the authors provide evidence showing that miR-194-3 targets CHD4 and that treatment with miR-194-3 has similar effects as silencing CHD4, there is no evidence to show that miR-194-3 affects cell proliferation and apoptosis by targeting CHD4. The authors should consider conducting an experiment cotreating granulosa cells with miR-194-3 mimic and si-CHD4 to investigate the pathway and provide more evidence for this pathway. More specific comments can be found below.

Specific Comments:

L40-42: “In summary, miR-194-3 regulated the proliferation and apoptosis of Zhedong white goose follicular GCs by directly targeting CHD4”. This is speculative as there is no evidence provided that miR-194-3 has these effects by directly interacting with CHD4

L48: this sentence is confusing. Consider omitting “strict screening” to make this sentence clearer

L87-89: the link to Sifrim-Hitz-Weiss syndrome does not seem relevant

L90-91: This sentence is unclear, consider omitting “is warranted” to improve

L114-115: The ovary is a heterogeneous tissue and sampling methods could greatly influence results. Please provide more detail (location of collection, quantity, any removal of follicles, etc.) about ovarian tissue collection as the ovarian structure would greatly differ between brooding and laying hens and sampling could vary between birds within the laying phase.

L119-121: the follicle classification explanation is confusing- it may be clearer to state that follicles were first separated into preovulatory (F1-F5) and prehierarchal follicles (0-10 mm) and prehierarchal follicles were further classified by size and color: small white follicle…

L121-122: were granulosa cells collected from individual birds or pooled?

L123: indicates that cells were seeded in 6-well plates, however, some experiments use different sized wells. The methods for granulosa cell culture should be rewritten to reflect any differences in experiments.

L132-136: how were these vectors designed?

L135: specify the name for CHD4 interference vector as seen in figures (si-CHD4)

L137: which genome assembly was used for CHD4?

L142-144: the concentrations tested for miR-194-3 mimic and inhibitor should be presented in the methods and the results of the optimal working concentrations should be discussed in the results section

L145: “cells were harvested at 48h after transfection”- this is inconsistent with protein analysis methods where cells were harvested after 72 h

L152: what machine was used for RT-qPCR?

L152-155: concentrations for RT-qPCR reactions (primers, SYBR, cDNA) should be included rather than volumes

L158: “three duplications” should be changed to triplicates if 3 of the same reaction was run per sample

L162: Table 2 has inconsistencies in formatting

L174: are the three biological replicates equivalent to cells from three separate geese?

L178: were there technical replicates for this assay?

L192: specify amount of protein used

L198: specify the catalogue number used for each antibody

L206: were there technical replicates for this assay?

L214: were there technical replicates for this assay?

L228: describe Hoechst staining protocol

L234: “All date” should be “all data”

L234-235: western blot analysis would be better suited in the western blot section of the methods

L237: A one way ANOVA is not appropriate for comparisons involving fewer than 3 groups. The statistical analysis section should be revised to clearly state which statistical tests were used for each comparison. This section should also include information about the software and packages used to run the analyses to ensure transparency and reproducibility.

Line 249-251: this should be in the discussion section

Line 255: the authors should clarify that the notation “a,b,c p<0.05” indicates that means labeled with different letters are significantly different from one another based on post hoc tukey analysis.

L257: the methods for this analysis is not presented in the methods section

L263: the specific genes showing this pattern should be listed in the text

L269-270: the interpretation of CDH4 and miR-194-3 should be moved to the discussion section.

L278: the interpretation of the luciferase activity assay should be moved to the discussion section

L281: Please include the species used for this analysis in the figure legend

Figure 2E: the y axis should specify mRNA expression

Figure 2I: This panel is not described in the figure legend

L303: this statement is more of an interpretation of the results rather than a description of the results, please revise to reflect a description of the cell populations

L312-313: this statement is more of an interpretation of the results rather than a description of the results, please revise to reflect a description of the cell populations

L315: “Collectively, these results …” should be moved to the discussion section

L319-320: list concentrations tested

Figure 3A: y axis should specify mRNA expression

Figure 3D: staining is labeled as DAPI in the figure, however, the figure legend and methods describe Hoechst staining

Figure 3E: a better description of this panel in the legend would improve readability

Figure 3H and 3I: treatment group labels for mimic should be consistent with               labels from other panels

Figure 4A: y axis should specify mRNA expression

Figure 4D: staining is labeled as DAPI in the figure, however, the figure legend and methods describe Hoechst staining

Figure 3H and 3I: treatment group labels for inhibitor should be consistent with labels from other panels

L331: list concentrations tested

L350: was Caspase-3 antibody against total caspase or cleaved caspase. The argument for increased apoptosis rate would be strengthened by including cleaved caspase-3 to show activation

L 351: how was apoptotic rate calculated?

L357-359: “These findings demonstrated …” should be moved to the discussion

Figure 5: labels for miR-194-3 mimic and inhibitor should be consistent across figure

L393-395: “Collectively, these findings…” should be moved to the discussion section

L466-469: “This approach aligned with current …” this sentence is not relevant to the purpose of this study.

Comments on the Quality of English Language

The quality of the English can be improved in a few sections of the text. There are some grammatical errors that require revision. 

Reviewer 2 Report

Comments and Suggestions for Authors

Overall a well designed and sound study of importance. My concerns are regarding the data and its interpretation. 

  1. Why do authors use a 72h time point for the western blot but resort to 48h for most of their other findings? It is important to assay at the same time point to capture physiologically similar events
  2. Authors are asked to share full western blot images as supplementary data. No cropping, no editing. This will allow to determine if the indicated bands are relevant or not and if they are at the correct size
  3. Show data to indicate level of knockdown brought about by the siRNAs. Also, why do the authors only use one siRNA? Did they check more? If so, the data should be shared.
Comments on the Quality of English Language

The article requires language editing - multiple issues with grammar, tense, and typos

Reviewer 3 Report

Comments and Suggestions for Authors

This study investigated the regulatory role of miR-194-3 and its target genes on the proliferation and apoptosis of goose granulosa cells. The following revisions are suggested:

Standardize numerical notation (use Arabic numerals or English words consistently, e.g., line 111).

Replace "stomach" with specific terms such as gizzard or proventriculus.

Clarify the composition of "ovarian tissue" throughout the text (e.g., ovarian stroma, follicles, or whole ovary).

Specify the developmental stage of granulosa cells used in each experiment.

Result 1 shows the highest miR-194-3 expression in F1 follicles. Does this imply miR-194-3 primarily participates in ovulation?

Result 1 indicates peak miR-194-3 expression in F1 follicles. Why were prehierarchical follicles ultimately selected for follow-up studies? Explain the connection between Result 1 and subsequent experiments.

The conclusion should clearly indicate which stage of follicular granulosa cells miR-194-3/CHD4 acts on.

Round 2

Reviewer 1 Report

Comments and Suggestions for Authors

               The authors have made significant efforts to address previous comments and their efforts have improved the quality of the manuscript. While the authors addressed most of the smaller concerns, several of the larger concerns were not addressed. The authors were able to clarify some of the methodology and reorganize some of the results section to exclude interpretive statements, however, the manuscript could benefit from more restructuring and more details in the methodology to further improve the readability and reproducibility. For example, the introduction includes a summary of the findings, which would more appropriately be placed in the abstract or discussion. Some sections of the methods could also be improved to ensure reproducibility.

               The experimental design and data interpretation presented in this manuscript could be improved. The experiments performed show that miR-194-3 is a target of CHD4 and that overexpression and CHD4 knockdown have similar effects on cell proliferation and apoptosis in granulosa cells of F1 follicles. The authors conclude that miR-194-3 inhibits cell proliferation and induces apoptosis through CHD4 downregulation. While efforts were made in this version of the manuscript to avoid overstating this conclusion, some areas still imply that miR-194-3 works directly through CHD4 (ex: L14-17, L535-536). While this may be possible given the results of this paper, no rescue experiment was performed to show that the effects of miR-194-3 are dependent on CHD4. The authors present several other possible targets of miR-194-3 which exhibited similar expression patterns as CHD4 with miR-194-3 overexpression and inhibition. Investigating the expression pattern of CHD4 across follicle development alongside the expression pattern of miR-194-3 would also be important to show their relationship. More experimentation is required to show that miR-194-3 affects granulosa cell function by directly targeting CHD4.

The authors were able to clarify the ovarian sampling methods used in this version, although it remains unclear how follicles were sampled (RT-qPCR of whole follicles or granulosa cells?, were all follicles in a follicle stage pooled?). This sampling method may not be appropriate to evaluate molecular differences between geese in the brooding and laying ovaries. As previously mentioned, ovaries of geese in the brooding phase are likely regressed and have different numbers of follicles <10 mm due to the change in endocrine environment compared to ovaries of laying geese. Therefore, differences in gene expression in ovaries of geese in the brooding vs laying phase may be due to the differences in follicle number. A more conservative ovarian sampling method may be necessary to compare birds in these two phases. The authors should consider measuring miR-194-3 and CHD4 in granulosa cells of ovaries of these two phases if follicles are present in brooding geese. A figure showing pictures of the ovaries sampled to show the sizes of follicles present could benefit the paper as well.

While the authors made progress in editing this manuscript, further experimentation is required to support the authors’ claims.

Comments on the Quality of English Language

There are some grammatical errors throughout the text which require attention. 

Author Response

Thank you very much for your valuable feedback. Please refer to the Word document for the specific revisions.

Reviewer 2 Report

Comments and Suggestions for Authors

The authors have addressed all of the concerns raised

Author Response

Thank you very much for your valuable feedback.